# Store-Operated Ca^2+^ Entry as a Putative Target of Flecainide for the Treatment of Arrhythmogenic Cardiomyopathy

**DOI:** 10.3390/jcm12165295

**Published:** 2023-08-14

**Authors:** Francesco Moccia, Valentina Brunetti, Teresa Soda, Pawan Faris, Giorgia Scarpellino, Roberto Berra-Romani

**Affiliations:** 1Department of Biology and Biotechnology “Lazzaro Spallanzani”, University of Pavia, 27100 Pavia, Italy; valentina.brunetti01@unipv.it (V.B.); giorgia.scarpellino@unipv.it (G.S.); 2Department of Health Sciences, University of Magna Graecia, 88100 Catanzaro, Italy; teresa.soda@unicz.it; 3Department of Brain and Behavioral Sciences, University of Pavia, 27100 Pavia, Italy; faris.pawan@unipv.it; 4Department of Biomedicine, School of Medicine, Benemérita Universidad Autónoma de Puebla, Puebla 72410, Mexico; rberra001@hotmail.com

**Keywords:** flecainide, arrhythmogenic cardiomyopathy, cardiac mesenchymal stromal cells, fibro-adipogenic differentiation, Ca^2+^ oscillations, store-operated Ca^2+^ entry, STIM1, Orai1

## Abstract

Arrhythmogenic cardiomyopathy (ACM) is a genetic disorder that may lead patients to sudden cell death through the occurrence of ventricular arrhythmias. ACM is characterised by the progressive substitution of cardiomyocytes with fibrofatty scar tissue that predisposes the heart to life-threatening arrhythmic events. Cardiac mesenchymal stromal cells (C-MSCs) contribute to the ACM by differentiating into fibroblasts and adipocytes, thereby supporting aberrant remodelling of the cardiac structure. Flecainide is an I_c_ antiarrhythmic drug that can be administered in combination with β-adrenergic blockers to treat ACM due to its ability to target both Na_v_1.5 and type 2 ryanodine receptors (RyR2). However, a recent study showed that flecainide may also prevent fibro-adipogenic differentiation by inhibiting store-operated Ca^2+^ entry (SOCE) and thereby suppressing spontaneous Ca^2+^ oscillations in C-MSCs isolated from human ACM patients (ACM C-hMSCs). Herein, we briefly survey ACM pathogenesis and therapies and then recapitulate the main molecular mechanisms targeted by flecainide to mitigate arrhythmic events, including Na_v_1.5 and RyR2. Subsequently, we describe the role of spontaneous Ca^2+^ oscillations in determining MSC fate. Next, we discuss recent work showing that spontaneous Ca^2+^ oscillations in ACM C-hMSCs are accelerated to stimulate their fibro-adipogenic differentiation. Finally, we describe the evidence that flecainide suppresses spontaneous Ca^2+^ oscillations and fibro-adipogenic differentiation in ACM C-hMSCs by inhibiting constitutive SOCE.

## 1. Introduction

Arrhythmogenic cardiomyopathy (ACM) is a genetic disease that leads to the progressive replacement of ventricular myocardium with fibrofatty scar tissue, thereby predisposing the patient to ventricular arrhythmias and sudden cardiac death (SCD) [1,2]. ACM may affect the right ventricle (arrhythmogenic right ventricular cardiomyopathy, ARVC), the left ventricle (arrhythmogenic left ventricular cardiomyopathy, ALVC), or both cardiac ventricles and is the main cause of SCD in young people and athletes [1,2]. A crucial role in the pathogenic mechanism of ACM can be played by the fibro-adipogenic differentiation of cardiac mesenchymal stromal cells (C-MSCs), which renders the cardiac tissue more likely to develop the arrhythmic events leading to the patient’s death [2,3]. Intriguingly, a recent investigation showed that the I_c_ antiarrhythmic drug (AAD) flecainide inhibited fibro-adipose differentiation in human C-MSCs (C-hMSCs) expanded from ACM patients (ACM C-hMSCs) by blocking store-operated Ca^2+^ entry (SOCE) [4], which represents the most important Ca^2+^ entry pathway in non-excitable cells [5,6,7], including MSCs [8,9,10]. Herein, we first describe the genetic background and therapeutic options of ACM. Then, we briefly recapitulate the main molecular mechanisms targeted by flecainide to mitigate arrhythmic events. Subsequently, we describe the role of intracellular Ca^2+^ oscillations in determining MSC fate. Next, we discuss our recent work showing that spontaneous Ca^2+^ oscillations in ACM C-hMSCs are accelerated to stimulate their fibro-adipogenic differentiation. Finally, we describe the evidence that flecainide suppresses spontaneous Ca^2+^ oscillations and fibro-adipogenic differentiation in ACM C-hMSCs by inhibiting constitutive SOCE and interfering with endoplasmic reticulum (ER) Ca^2+^ refilling.

## 2. Pathological Background and Treatment of ACM

Recently, an evidence-based reevaluation of published ARVC, the most common form of ACM, genes was carried out by experts in the field and showed that only a small number of genes encoding for desmosomal proteins, such as plakophilin-2 (*PKP2*), plakoglobin (*JUP*), and desmoglein 2 (*DSG2*), were definitively or moderately associated with ARVC (Table 1) [11]. Other genes that encode adherens junction proteins, such as cadherin 2 (*CDH2*) or catenin α3 (*CTNNA3*), that are also associated with ACM (Table 1) [1,2], only showed a moderate association with ACM [11]. Mutations in genes encoding proteins that are unrelated to intercellular junctions, such as cytoskeletal proteins, ion channels, and transporters, were also reported (Table 1) [1,2]; however, they were also shown to be weakly associated with ARVC [11]. This Gene Curation Expert Panel refuted the gene encoding for type 2 ryanodine receptor (*RyR2*), which has long been associated with ACM [1,2], as an ARVC gene [11].

Besides maintaining the mechanical stability of the heart and enabling the electrical coupling between ventricular cardiomyocytes, desmosomes may also play a crucial role in intracellular signal transduction and gene expression [2]. For instance, when the desmosomal assembly is disrupted by genetically defective proteins, plakoglobin 2 translocates from the intercalated discs to the nucleus, where it may compete with β-catenin to suppress the Wnt/β-catenin signalling pathway and elicit a gene transcriptional switch from myogenesis to fibrogenesis and adipogenesis [1,2,12]. A series of recent studies demonstrated that desmosomal mutations may also affect c-MSCs [13,14], which support the structural and functional integrity of the myocardium [15,16]. C-MSCs represent the primary source of myofibroblasts and adipocytes in the heart of ACM patients [13,14]. The fibrofatty replacement of ventricular myocardium may favour life-threatening arrhythmias, such as ventricular fibrillation and sustained ventricular tachycardia (VT), that ultimately cause SCD [2,3]. Patients who develop symptoms or have been diagnosed with ACM after a genetic screening or an electrocardiogram can be treated with implantable cardioverter-defibrillators (ICDs), radiofrequency catheter ablation, heart transplantation, β-blockers, or AADs [3,17]. Currently, ICDs represent the only therapeutic approach that can effectively reduce patient mortality, although they can be recommended only in the presence of clear arrhythmic risks, such as ventricular fibrillation and sustained VT [17]. AADs can be administered to reduce the burden of ACM-associated arrhythmias in symptomatic patients with non-sustained VT; moreover, AADs can be recommended as adjunctive therapy to mitigate the morbidities associated with ICDs (e.g., frequent device discharges and VT recurrences) and to facilitate catheter ablation [3,17]. Sotalol and amiodarone represent the first-line AADs in clinical practice, alone or in conjunction with β-blockers; however, there is no evidence that AAD therapy is effective at protecting patients from SCD. Furthermore, long-term treatment with amiodarone could induce extracardiac toxicity, so its use should be balanced in terms of effective arrhythmia suppression or reduction [3,17]. New hope for the pharmacological treatment of ACM has been sparked by the class I_c_ AAD flecainide ((RS)-N-(piperidin-2-ylmethyl)-2,5-bis(2,2,2-trifluoroethoxy) benzamide); C_17_H_20_F_6_N_2_O_3_) [18,19,20,21], which was approved by the Food and Drug Administration (FDA) in 1984 for the treatment of symptomatic sustained VT [18,20,22]. Flecainide targets a panel of ion channels and transporters that shape the cardiac action potential (AP; e.g., Na_v_1.5, K_v_1.5, and K_v_11.1) or the depolarization-induced Ca^2+^ transient responsible for cardiomyocyte contraction (e.g., type 2 ryanodine receptor, or RyR2) [22,23]. In addition, a recent investigation showed that flecainide may also inhibit fibro-adipose differentiation in ACM C-hMSCs by targeting SOCE [4].

**Table 1 jcm-12-05295-t001:** Genes whose mutations are associated with ACM.

Gene	Encoded Protein	Estimated Frequency (%)	Chromosomal Location
Desmosomes			
PKP2	Plakophilin-2	19–46	12p11.21
DSP	Desmoplakin	1–16	6p24.3
DSG2	Desmoglein-2	2.5–10	18q12.1
DSC2	Desmocollin-2	1–8	18q12.1
JUP	Junction plakoglobin	Rare	17q21.2
Adherens junctions			
CTNNA3	Catenin-α3	Rare	10q21.3
CDH2	Cadherin 2	Rare	18q12.1
Cytoskeleton			
LMNA	Lamin A/C	Rare	1q22
DES	Desmin	Rare	2q35
FLNC	Filamin C	Rare	7q32.1
TTN	Titin	Rare	2q31.2
Ion transport			
SCN5A	Na_V_1.5	Rare	3p22.2
PLN	Phospholamban	Rare	6q22.31

For references, see: [1,17].

## 3. Anti-Arrhythmic Effects of Flecainide: From Na_v_1.5 to RyR2

A number of excellent reviews have provided a comprehensive description of the molecular pharmacology of flecainide [22,23,24]. Briefly, flecainide was first synthesised in 1972 by incorporating fluorine in local anesthetics, thereby increasing their stability, and it is nowadays recommended among the first-line therapeutic options to manage arrhythmias in patients without severe hemodynamic stability or significant structural heart disorders [25]. Flecainide reduces that fast inward Na^+^ current responsible for phase 0 of the cardiac AP by inhibiting the underlying Na_v_1.5 channel protein (Figure 1) [22,26,27]. In addition, flecainide may inhibit the late Na^+^ current (I_NaL_) [26], which can arise because of recovery from inactivation and late re-opening of Na_v_1.5 channels during the sustained plateau in Purkinje fibres [28].

Flecainide may also inhibit the rapid delayed rectifier K^+^ current (I_KR_), which is carried by K_v_11.1 channels (human Ether-à-go-go-Related Gene, or hERG) and is responsible for the late repolarization restoring the resting membrane potential in cardiomyocytes (Figure 2) [26,30]. Flecainide can further block the fast transient outward K^+^ current (I_tof_) (Figure 2), which is mediated by K_v_4.2 channels and contributes to the early depolarization phase, and the ultrarapid K^+^ current (I_KUR_) (Figure 2), which is carried by K_v_1.5 channels and is the major delayed rectifier K^+^ current in the atria [22,26].

At concentrations higher than 10 µM, flecainide inhibits K_v_4.2, which mediates the fast transient outward K^+^ current (I_tof_), thereby potentially increasing APD in atrial and ventricular cardiomyocytes. In accordance with this, this inhibitory effect is exerted at supra-clinical doses of flecainide (IC_50_ = 15.2 µM) and, therefore, it is not expected to occur in the patients. Finally, flecainide may also inhibit K_v_1.5 (IC_50_ = 237.1 µM), which mediates the ultrarapid delayed rectifier K^+^ current (IK_UR_) in the atria, where it is particularly abundant, and in the ventricles, thereby increasing the APD. This effect also occurs at supra-clinical doses and is unlikely to occur in patients.

Globally, flecainide prolongs the AP duration (APD) and AP refractoriness in ventricular and atrial cardiomyocytes, while they are both shortened in Purkinje fibres due to Na_v_1.5 channel blockade [22,26]. However, in human atrial cardiomyocytes, the flecainide-induced increase in APD and refractoriness is more evident when the AP presents a long-lasting plateau preceded by a notch [26]. Intriguingly, the kinetics of flecainide unbinding from the Na_v_1.5 channel pore during diastole are quite slow, thereby prolonging the refractoriness more than the APD (a phenomenon known as post-repolarization refractoriness), reducing excitability, and slowing intracardiac conduction, even at physiological heart rates, throughout all cardiac tissues [26]. As discussed in [20,26,31], the clinical concentration range of flecainide may selectively inhibit I_Na_ and I_KR_. Therefore, Na_v_1.5 and K_v_11.1 channels represent the main molecular targets of flecainide in arrhythmic patients. Flecainide can indeed be used to prevent and treat various types of arrhythmias, such as paroxysmal atrial fibrillation, supraventricular tachycardia, and arrhythmic long QT syndromes (LQTS) [18,20,22]. Flecainide is particularly suitable for LQTS (LQTS3) type 3, which is associated with gain-of-function mutations in Na_v_1.5 channels (Figure 1) [20,22], which increase I_NaL_ [28,29]. Recent studies further showed that flecainide can also block RyR2, thereby inhibiting spontaneous RyR2-mediated sarcoplasmic reticulum (SR) Ca^2+^ release, which leads to delayed afterdepolarization (DADs) and triggered activity (Figure 1). Spontaneous mobilisation of SR Ca^2+^ can indeed be compensated by the electrogenic Na^+^/Ca^2+^ exchanger (NCX), which exports 1 Ca^2+^ out for 3 Na^+^ in during each cycle (Figure 1) and, while doing so, brings about an inward, depolarizing current [22,23]. Therefore, flecainide can also be used to treat catecholaminergic polymorphic ventricular tachycardia (CPVT) (Figure 1), a genetic arrhythmogenic disorder associated with mutations in RyR2 (CPVT1) or in the SR Ca^2+^-binding protein calsequestrin (CPVT2), which both lead to dysregulated SR Ca^2+^ release through RyR2 in response to β-adrenergic stimulation [23,32,33]. Flecainide binds to multiple sites within the RyR2 channel pore in a voltage-dependent manner but may also interact with several cytoplasmic sites in a voltage-independent manner [23,33]. The IC_50_ of flecainide-induced inhibition of RyR2 in transgenic mouse models of CPVT is ~2 µM [32,34], i.e., in the clinical therapeutic range of flecainide.

The use of flecainide has been prohibited in patients with myocardial infarction, left ventricular dysfunction, or structural heart disease based upon the increase in their mortality upon treatment with flecainide registered during the Cardiac Arrhythmia Suppression Trial (CAST) study [35]. The CAST study reported a significant increase in the mortality of patients treated with I_c_ AADs, including flecainide, as compared to subjects treated with placebo, with the primary causes of death being arrhythmias [35]. Nevertheless, a series of recent investigations [18,20,36,37,38], as well as a retrospective analysis by Rolland et al. [19], confirmed that flecainide is both effective and safe for reducing atrial fibrillation and symptomatic ventricular arrhythmias in patients with different structural heart disorders, including coronary artery disease (CAD). Lavalle and coworkers recently affirmed that the therapeutic use of flecainide is strongly limited by the anachronistic interpretation of the CAST since the introduction of more efficient diagnostic tools, such as cardiac magnetic resonance imaging (CMR) rather than transthoracic echocardiography, will be helpful to identify the type of structural heart disorder that may really prevent flecainide treatment [20]. In addition, it has been put forward that thrombolysis, as well as primary percutaneous coronary intervention, may enable the selection of CAD patients who do not present myocardial scar and residual ischaemia and for whom there is no straightforward evidence against the prescription of flecainide [20].

## 4. The Rationale for Using Flecainide to Treat Ca^2+^-Dependent Ventricular Arrhythmias in ACM

A consensus statement published by the Heart Rhythm Society (HRS) in conjunction with several organisations, including the American Heart Association (AHA), American College of Cardiology (ACC), and European Heart Rhythm Association (EHRA), recommended the use of flecainide in combination with β-blockers to treat ACM patients refractory to single-agent therapy or catheter ablation [19,21,39,40]. A pilot randomised clinical trial is currently evaluating the effectiveness of flecainide in reducing ventricular arrhythmia in ACM patients (Pilot Randomized Trial With Flecainide in ARVC Patients, NCT03685149, currently ongoing). Blockade of Na_v_1.5 channels represents the most obvious mechanism to explain the anti-arrhythmic effects exerted by flecainide in ACM [26,38]. Emerging evidence, however, suggests that dysregulated RyR2-mediated SR Ca^2+^ release can also generate pro-arrhythmic Ca^2+^ events in ACM [41]. By using a cardiomyocyte-specific *PKP2* knockout mouse model, Cerrone et al. found that PKP2 deletion leads to a complex remodelling of the Ca^2+^ handling machinery, involving a reduction in the SR Ca^2+^ leakage caused by the downregulation of calsequestrin and RyR2 proteins [42]. This causes an elevation in SR Ca^2+^ content, which increases the amplitude and duration of spontaneous RyR2-mediated SR Ca^2+^ release because of the increase in intraluminal Ca^2+^ concentration. As a consequence, PKP2 deletion rendered RyR2 more prone to release Ca^2+^ during excitation-contraction coupling and to generate pro-arrhythmic Ca^2+^ release events during β-adrenergic stimulation [42]. A follow-up investigation confirmed that, in right ventricle-derived *PKP2*-deficient cardiomyocytes, the frequency of Ca^2+^ sparks, i.e., the elementary events of SR Ca^2+^ release [43], and of pro-arrhythmic afterdepolarizations increased before the onset of the cardiomyopathy due to an increase in SR Ca^2+^ content [44]. Finally, the same group showed that physical exercise caused a dramatic increase in the frequency and amplitude of SR Ca^2+^ sparks in *PKP2*-deficient mouse hearts because of the hyperphosphorylation of phospholamban, which accelerates Ca^2+^ uptake into the SR via the SarcoEndoplasmic Reticulum Ca^2+^-ATPase, during β-adrenergic stimulation [45]. Interestingly, blocking RyR2 with flecainide reduced the arrhythmia burden induced by β-adrenergic stimulation in *PKP2*-knockout mice [42]. Furthermore, a recent report confirmed that the Ca^2+^ cycling machinery was also altered in homozygous *Dsg2* mutant mice (*Dsg2*^mut/mut^) [46]. Cytosolic Ca^2+^ overload during endurance exercise stimulates calpain-1 association with the mitochondria, thereby leading to the cleavage of mitochondrial-bound apoptosis-inducing factor (AIF). The truncated AIF translocated to the nucleus and triggered large-scale DNA fragmentation and cell damage, an effect that was exacerbated by mitochondrial-driven AIF oxidation. This induced myocyte necrosis in the heart of exercised *Dsg2*^mut/mut^ mice [46].

On the one hand, these findings confirm that, as reported in other arrhythmic disorders (e.g., CPVT, Timothy syndrome, Brugada syndrome, and long and short QT syndrome), ACM is also associated with the deregulation of the Ca^2+^ cycling machinery in cardiomyocytes [47,48]. On the other hand, they support the notion that flecainide could effectively treat ACM patients by targeting the Ca^2+^ signalling machinery. In accordance with this hypothesis, flecainide decreased the frequency of Ca^2+^ sparks and normalised the Ca^2+^ transients elicited by membrane depolarization in inducible pluripotent stem cell-derived cardiomyocytes isolated from an ACM patient bearing a mutation in the *DSC2* gene [49]. It has long been known that oscillations in intracellular Ca^2+^ concentration ([Ca^2+^]_i_) regulate MSC differentiation into a variety of cells belonging to multiple lineages [50,51,52]. Moreover, a recent study carried out in our laboratory showed that intracellular Ca^2+^ oscillations drive proliferation in C-hMSCs [53]. In the next paragraphs, we first discuss the mechanisms underpinning intracellular Ca^2+^ oscillations in MSCs. Then, we illustrate how an increase in the frequency and amplitude of spontaneous Ca^2+^ activity drives fibro-adipose remodelling in ACM C-MSCs. Finally, we describe how flecainide inhibits spontaneous Ca^2+^ waves and fibro-adipogenic differentiation by targeting SOCE in ACM C-hMSCs.

## 5. Intracellular Ca^2+^ Oscillations Regulate MSC fate

The signalling mechanisms and functions of intracellular Ca^2+^ oscillations have been primarily characterised in bone marrow (BM)-derived MSCs. A landmark series of studies conducted by Kawano et al. showed that, in BM-derived human MSCs (BM-hMSCs), spontaneous Ca^2+^ oscillations were driven by ATP autocrine/paracrine signalling [54,55,56]. ATP stimulated the Gq-protein coupled P2Y1 receptor to recruit phospholipase Cβ (PLCβ), which cleaved phosphatidylinositol-4,5-bisphosphate (PIP_2_), a minor phospholipid component of the plasma membrane, into diacylglycerol and inositol-1,4,5-trisphosphate (InsP_3_). InsP_3_ triggers rhythmic Ca^2+^ release from the endoplasmic reticulum (ER), the most abundant intracellular Ca^2+^ reservoir, by gating the so-called InsP_3_ receptors (InsP_3_Rs) [54,55,56]. These are Ca^2+^-permeable, non-selective cation channels that are embedded within ER cisternae and whose opening can be finely tuned by cytosolic Ca^2+^ levels [57,58]. BM-hMSCs express type 1 and type 2 InsP_3_Rs (InsP_3_R1 and InsP_3_R2, respectively), which are the most suitable InsP_3_Rs to generate rhythmic Ca^2+^ spikes in response to the continuous production of InsP_3_ because of their sensitivity to InsP_3_ and Ca^2+^ [56,59]. Intriguingly, InsP_3_-induced spontaneous Ca^2+^ spikes in BM-hMSCs fade away in the absence of extracellular Ca^2+^; therefore, Ca^2+^ influx is required to refill the ER Ca^2+^ pool and support the rhythmic Ca^2+^ activity over time [54], as widely reported in other cell types [60,61,62,63,64]. Electrophysiological recordings revealed that BM-hMSCs do not present voltage-gated Ca^2+^ currents, although transcripts encoding for the Cav1.1 and Cav3.2 α subunits of voltage-operated Ca^2+^ channels were detected. Conversely, these cells express a measurable inwardly-rectifying store-operated Ca^2+^ current [54], also known as I_CRAC_ (Ca^2+^ release-activated Ca^2+^ current) (Figure 3). Intriguingly, by recharging the ER Ca^2+^ store, SOCE could also sensitise InsP_3_Rs via an increase in intraluminal Ca^2+^, which leads to periodic InsP_3_R activation and enhances the frequency of InsP_3_Rs-mediated Ca^2+^ release [57,58,65].

The I_CRAC_ is the primary Ca^2+^-entry pathway sustaining intracellular Ca^2+^ oscillations over time in non-excitable cells and mediates an influx of Ca^2+^ that is commonly termed store-operated Ca^2+^ entry (SOCE) [5,6,57,58]. SOCE is activated whenever the ER Ca^2+^ concentration ([Ca^2+^]_ER_) in proximity of discrete sub-compartments of the peripheral ER falls below a threshold concentration because of ER Ca^2+^ release through InsP_3_Rs [66,67]. SOCE is mediated by the physical interaction between two ubiquitously expressed proteins (Figure 3): STIM, which is the sensor of [Ca^2+^]_ER_, and Orai, which contributes the Ca^2+^-selective channel protein on the plasma membrane [5,6,68]. STIM presents two isoforms, i.e., STIM1 and STIM2, whereas Orai displays three paralogues, i.e., Orai1, Orai2, and Orai3. Briefly, a reduction in [Ca^2+^]_ER_ stimulates STIM proteins to assemble into oligomers that redistribute to peripheral ER-plasma membrane junctions, known as puncta, while extending their cytosolic COOH-termini, which contain the STIM Orai-activating region/CRAC-activating domain (SOAR/CAD), towards the inner leaflet of the plasma membrane (Figure 3). The SOAR/CAD domain, in turn, binds to and gates the hexameric Orai channels, thereby activating the I_CRAC_ and inducing SOCE (Figure 3) [5,6,69]. STIM2 presents a lower Ca^2+^ affinity as compared to STIM1 and is, therefore, activated upon a modest fall in [Ca^2+^]_ER_. It turns out that STIM2 is regarded as the most suitable isoform to tonically activate SOCE even in the absence of extracellular stimulation and to maintain ER Ca^2+^ levels [70]. Conversely, STIM1 is activated after a significant depletion of [Ca^2+^]_ER_ during InsP_3_-induced Ca^2+^ release and sustains the Ca^2+^ response to extracellular cues [5,6]. Nevertheless, STIM2 can sustain SOCE elicited by agonist-induced sub-threshold depletion of the ER Ca^2+^ content [68,71] as well as support STIM1-dependent recruitment to the ER-plasma membrane during supra-threshold stimulation [68,72]. In addition, STIM1 can be activated even in the absence of an extracellular agonist when the resting [Ca^2+^]_ER_ is too low to prevent its constitutive activation [73,74,75]. Orai1 has long been regarded as the primary pore-forming subunit of CRAC channels [5,69]. However, recent evidence suggests that Orai2 and/or Orai3 can assemble into heteromultimeric channels with Orai1 and serve as negative modulators of the I_CRAC_ in naïve cells [76,77,78]. The studies conducted by Kawano et al. date back to the early decade of this century [54,55,56], when STIM and Orai were not known to mediate the I_CRAC_. Therefore, we still do not know whether they support SOCE in BM-hMSCs. Nevertheless, genetic silencing of STIM1, Orai1, and/or Orai3 reduced SOCE in mouse BM-derived MSCs (BM-mMSCs) [79,80,81] and in human dental pulp-derived MSCs (DP-hMSCs) [82].

Spontaneous intracellular Ca^2+^ oscillations have been described not only in BM-hMSCs [54,83,84] but also also in human adipose tissue-derived MSCs (AD-hMSCs) [51,85]. Intracellular Ca^2+^ oscillations regulate stem cell differentiation by stimulating the expression of genes that control cellular fate and silencing those that are required for self-renewal [52,86,87,88,89,90]. The spatio-temporal profile of spontaneous Ca^2+^ activity can selectively change during hMSC differentiation in tissue-specific lineages by showing either an increase or a decrease in the spike frequency [52]. Spontaneous Ca^2+^ oscillations disappear during hMSC differentiation towards an adipogenic phenotype both when they derive from the BM [56,91] and from the adipose tissue [51]. The loss of spontaneous Ca^2+^ activity reduces the nuclear translocation of the Ca^2+^-sensitive transcription factor, nuclear factor of activated T-cells (NFAT), which is likely to prevent hMSC differentiation [56]. Conversely, BM-hMSCs undergoing neuronal differentiation show an increase in the frequency of spontaneous Ca^2+^ spikes, which become quite irregular in terms of amplitude and shape [90,92]. A more robust Ca^2+^ spiking could govern BM-hMSC differentiation toward a stable neuronal phenotype by promoting the phosphorylation of CREB (cAMP response element binding protein) [93], a transcription factor that plays a crucial role in neurogenesis and can be activated by the Ca^2+^/CaM-dependent protein kinase IV [94]. The requirement of spontaneous Ca^2+^ oscillations for hMSC commitment to a specific cellular lineage is so tight that the intracellular Ca^2+^ activity in BM-hMSCs can be physically manipulated, for instance by electrostimulation [95], to favour osteogenic [91] or neuronal [96] differentiation. Similarly, the addition of inductive soluble factors, such as the InsP_3_-producing autacoid carbachol, has been successfully employed to increase the frequency of InsP_3_Rs-mediated spontaneous Ca^2+^ oscillations and thereby promote neuronal differentiation in AD-hMSCs [97]. As described above, SOCE is crucial to maintaining spontaneous Ca^2+^ oscillations in hMSCs; an increase or decrease in the rate of Ca^2+^ entry can differentially affect their differentiation outcomes. Therefore, it is not surprising that pharmacological manipulation of SOCE has been put forward as an alternative strategy to control hMSC differentiation [10,80].

## 6. Flecainide Inhibits SOCE and Prevents Ca^2+^-Dependent Fibro-Adipogenic Differentiation in ACM C-hMSCs

A recent investigation carried out by our laboratory has shown that intracellular Ca^2+^ signalling drives C-hMSC proliferation by stimulating extracellular signal-regulated kinase (ERK) phosphorylation [53]. In addition, this study confirmed that a functional SOCE is expressed in C-hMSCs and is sensitive to InsP_3_-induced depletion of the ER Ca^2+^ store [53]. Therefore, based upon the evidence provided by the Pompilio group that ACM C-hMSCs are more likely to undergo fibro-adipogenic differentiation as compared to C-hMSCs [13,14], we established a collaborative endeavour to understand whether spontaneous Ca^2+^ oscillations were dysregulated in ACM C-hMSCs. Then, we evaluated whether SOCE could be pharmacologically targeted to prevent aberrant lipid/fibrotic accumulation in the ACM heart [4].

### 6.1. Intracellular Ca^2+^ Oscillations Are Up-Regulated in ACM C-hMSCs

Single-cell imaging of the Ca^2+^-sensitive fluorophore, Fura-2, represents a widespread technique to analyse intracellular Ca^2+^ oscillations in tens to hundreds of non-excitable cells [62,63,64,98,99,100], including hMSCs [54,55,56]. This approach led to the observation that spontaneous Ca^2+^ oscillations in ACM C-hMSCs were significantly more robust than C-hMSCs in terms of percentage of oscillating cells, amplitude, and frequency (~2.2 mHz vs. ~1.2 mHz) of the single Ca^2+^ transients (Figure 4A,B) [4]. C-hMSCs are not amenable to genetic manipulation of the Ca^2+^ handling machinery [53]. However, a pharmacological approach confirmed that spontaneous Ca^2+^ oscillations in ACM C-hMSCs were driven by the Gq-protein-coupled P2Y1 receptors, triggered by rhythmic InsP_3_-induced ER Ca^2+^ release, and maintained over time by SOCE [4]. Conversely, RyR2 was not expressed, whereas voltage-gated Ca^2+^ entry did not support spontaneous Ca^2+^ oscillations [4]. Interestingly, the Mn^2+^ quenching assay, which is widely employed to monitor basal Ca^2+^ entry in non-excitable cells [70,75,101], revealed that SOCE was constitutively activated in C-hMSCs and was significantly enhanced in ACM C-hMSCs (Figure 4C,D) [4]. Constitutive SOCE regulates resting [Ca^2+^]_i_ and maintains [Ca^2+^]_ER_, thereby finely tuning the magnitude and duration of InsP_3_-induced ER Ca^2+^ release [70,73,74,75,101,102]. In accordance, basal [Ca^2+^]_i_ and InsP_3_-induced ER Ca^2+^ release were both up-regulated in ACM C-hMSCs [4]. These findings were supported by the molecular evidence that the expression of STIM1 and InsP_3_R2 was enhanced at both transcript and protein levels [4]. In addition, SERCA2B, which sequesters cytosolic Ca^2+^ into the ER lumen in non-excitable cells, was also up-regulated in ACM C-hMSCs [4]. Intriguingly, intracellular Ca^2+^ spiking was enhanced in C-hMSCs transduced with lentiviral particles containing a short hairpin against *PKP2*, which caused an increase in the expression levels of STIM1 and SERCA2B proteins [4].

These data support a model according to which (Figure 3): (1) STIM1 protein, the sensor of [Ca^2+^]_ER_ that is activated in response to large falls in [Ca^2+^]_ER_ occurring during autocrine/paracrine ATP signalling, is up-regulated in ACM C-hMSCs; (2) this leads to the recruitment of more Orai1 hexamers in the plasma membrane and therefore increases constitutive SOCE in ACM C-hMSCs; (3) the larger influx of Ca^2+^ causes a remarkable increase in [Ca^2+^]_ER_, also because of SERCA2B up-regulation, in ACM C-hMSCs; (4) the enhanced ER Ca^2+^ loading, on the one hand, sensitises InsP_3_Rs to open at a higher frequency; on the other hand, it increases the availability of free intraluminal Ca^2+^ that can be released during each Ca^2+^ spike. In this context, it is worth clarifying that, although the SOCE signal driving the spontaneous Ca^2+^ oscillations in C-hMSCs has been termed constitutive [4], it is stimulated by ATP released in either an autocrine or paracrine manner via P2Y1 receptor activation. The term “constitutive” reflects the lack of any agonist in the perfusate when intracellular Ca^2+^ oscillations and basal Ca^2+^ entry are monitored.

### 6.2. Intracellular Ca^2+^ Oscillations in ACM C-hMSCs: Is There Any Role for Lysosomal Ca^2+^ Mobilization?

It is worth noticing that ER Ca^2+^ release through InsP_3_Rs in C-hMSCs could also be activated by lysosomal Ca^2+^ release via two pore channels (TPCs) [53]. TPC1 and TPC2 are activated by the Ca^2+^-mobilizing second messenger, nicotinic acid adenine dinucleotide phosphate (NAADP) [103,104], and are both expressed in C-hMSCs, in which they trigger the intracellular Ca^2+^ oscillations by which foetal bovine serum stimulates proliferation [53]. Transmission electron microscopy revealed that lysosomal vesicles can establish quasi-synaptic membrane contact sites with the juxtaposed ER cisternae in C-hMSCs [53]. Herein, NAADP-induced lysosomal Ca^2+^ release via TPCs may activate InsP_3_Rs via the CICR process [53]. This signalling pathway may trigger long-lasting oscillations in [Ca^2+^]_i_ not only in C-hMSCs but also in many other cell types [62,64,105,106]. In addition, NAADP-induced lysosomal Ca^2+^ release may contribute to cardiac hypertrophy induced by β-adrenergic stimulation [107,108] and ischemia-reperfusion injury [109]. Therefore, future investigations will have to assess whether NAADP and TPCs play any role in the spontaneous Ca^2+^ oscillations arising in C-hMSCs and whether their contribution, if any, is increased in ACM. Intriguingly, SOCE has also been shown to support lysosomal Ca^2+^ refilling, such that the increase in constitutive SOCE has the potential to increase lysosomal Ca^2+^ release in ACM C-hMSCs [110].

### 6.3. Intracellular Ca^2+^ Oscillations Drive Fibro-Adipogenic Differentiation in ACM C-hMSCs

Unexpectedly, we found that spontaneous Ca^2+^ oscillations in ACM C-hMSCs showed an increase in amplitude and frequency (up to 8 mHz) during adipogenic differentiation [4]. This finding was surprising since, both in BM- and AD-derived MSCs, the Ca^2+^ spiking activity decreases when they are committed to an adipogenic phenotype [51,56]. The pharmacological blockade of the Ca^2+^ signalling machinery that sustains the accelerated Ca^2+^ spikes, i.e., SOCE and InsP_3_Rs, prevented ACM C-hMSCs from differentiating into either adipocytes or fibroblasts [4]. Therefore, it takes an increase in the amplitude and frequency of spontaneous Ca^2+^ oscillations to induce ACM C-hMSCs to undergo fibro-adipogenic differentiation. The multifunctional Ca^2+^/CaM-dependent protein kinase II (CaMKII) represents the most suitable decoder of intracellular Ca^2+^ oscillations not only in the brain but also in the heart [111,112,113]. CaMKII is a multimer of eight to twelve subunits that are present in four different isoforms (α, β, γ, and δ), each encoded by a distinct gene, arranged in two-stacked hexameric rings [114]. CaMKIIδ is the major isoform in the heart and exists as different spliced variants, e.g., CaMKIIδ_2/B,_ CaMKIIδ_2/C_, and CaMKIIδ_2/C_ [111]. Briefly, in response to an increase in [Ca^2+^]_i_, the Ca^2+^/CaM complex can bind to two neighbouring subunits, thereby enabling CaMKII to autophosphorylate on threonine 287 (Thr^287^). Autophosphorylation traps Ca^2+^/CaM and prolongs CaMKII activity even when the [Ca^2+^]_i_ returns to the baseline. Low-frequency Ca^2+^ transients are less effective than high-frequency Ca^2+^ transients to evoke autophosphorylation and Ca^2+^-independent activity since the Ca^2+^/CaM complex can dissociate from the holoenzyme between two consecutive Ca^2+^ spikes [115]. We found that CaMKIIδ and CaMKIIγ were the predominant CaMKII isoforms in C-hMSCs; however, there was no difference in CaMKII protein expression between C-hMSCs and ACM C-hMSCs. However, CaMKII protein in ACM C-hMSCs was hyperphosphorylated as compared to healthy cells, which is consistent with their greater eagerness to generate Ca^2+^ oscillations [4]. In agreement with this hypothesis, the pharmacological blockade of CaMKII activity with KN93 prevented fibro-adipogenic differentiation in ACM C-hMSCs [4]. Therefore, CaMKII is likely to translate the spontaneous Ca^2+^ oscillations that occur at a higher frequency in ACM C-hMSCs into a transcriptional programme that redirects their differentiation fate to fibroblasts or adipocytes. One could rightly notice that a frequency of 8 mHz might be too slow to enable the accumulation of Ca^2+^/CaM-bound CaMKII subunits sufficient for autophosphorylation and autonomous CaMKII activity [116]. Nevertheless, CaMKII does not only integrate the information encoded in the oscillation frequency but also in the length of the single Ca^2+^ pulses; an increase in the individual Ca^2+^ spike duration decreases the frequency threshold for CaMKII activation and autonomous activity [115]. The long duration of individual Ca^2+^ spikes in ACM C-hMSCs, ranging from ~200 s up to ~300 s (Figure 4B), enables coincident binding of CaM on neighbouring subunits and of Thr^287^ autophosphorylation even at 8 mHz [115,116]. For instance, the slow Ca^2+^ oscillations evoked by fertilisation in mouse oocytes were able to induce long-lasting oscillations in CaMKII activity that were crucial to resuming embryonic development [117]. The mechanism by which CaMKII stimulates the fibro-adipogenic differentiation of ACM-hMSCs is not clear [4]. However, CaMKII promoted adipogenesis in porcine BM-MSCs by increasing the expression of two adipogenic transcription factors, such as CCAAT/enhancer binding protein α (C/EBPα) and peroxisome proliferator activated receptor γ (PPARγ), by stimulating the PI3K/Akt-Fox01 signalling pathway [118].

### 6.4. Flecainide Abolishes Intracellular Ca^2+^ Oscillations and Fibro-Adipogenic Differentiation by Targeting SOCE

SOCE is either down-regulated or up-regulated in a growing number of disorders, including severe combined immunodeficiency [119], neurodegenerative diseases [120], genetic myopathies [121], pulmonary arterial hypertension [122], and cancer [123]. Preliminary evidence indicated that, even though Orai1 is weakly expressed in adult ventricular cardiomyocytes from humans and mice, its expression increases and leads to SOCE up-regulation in cardiac hypertrophy and heart failure [124]. Therefore, the pharmacological manipulation of SOCE with selective agonists or blockers of Orai1-mediated Ca^2+^ entry represents a promising therapeutic approach to rescue aberrant Ca^2+^-dependent processes in multiple cell types [65,122,125,126,127,128,129,130,131]. The evidence that flecainide may physically interact with many pro-arrhythmic ion channels led our group to hypothesise that it could also inhibit constitutive SOCE in ACM C-hMSCs. Consistently, single-cell Ca^2+^ imaging experiments carried out in our laboratory showed that 10 µM flecainide blocked constitutive SOCE (Figure 4E) and thereby inhibited InsP_3_-induced ER Ca^2+^ release in these cells [4]. As expected, based upon these findings, flecainide also suppressed spontaneous Ca^2+^ oscillations (Figure 4F) and interfered with ACM C-hMSC fibro-adipogenic differentiation [4]. Of note, ACM C-hMSC cells do not express Na_V_1.5; therefore, the inhibitory effect of flecainide is not related to the inhibition of voltage-gated inward Na^+^ currents [4]. In this view, it must be pointed out that Na_v_1.5-mediated Na^+^ currents could favour intracellular Ca^2+^ oscillations only by switching the NCX into the reverse (i.e., Ca^2+^ in, Na^+^ out) mode [132,133]. However, the pharmacological blockade of the reverse-mode NCX activity did not affect the spontaneous Ca^2+^ oscillations in ACM C-hMSCs [4]. It turns out that flecainide-induced SOCE inhibition underlies the suppression of both the unsolicited Ca^2+^ spikes and fibro-adipogenic differentiation that take place in C-hMSCs expanded from ACM patients. The mechanisms by which flecainide inhibits constitutive SOCE remain to be elucidated (Figure 3). SOCE in ACM C-hMSCs is also sensitive to two established inhibitors of this Ca^2+^-entry pathway, namely the pyrazole compounds Pyr6 and BTP-2 (YM-58483) [7,123,134,135,136,137]. These drugs are known to selectively block Orai1-mediated SOCE [7,123,134,135,136,137]; however, they could also affect Orai2 [138,139]. Nevertheless, Orai2 is a negative modulator of Orai1 [76,77,78]. Therefore, the direct inhibition of Orai2 is predicted to enhance, not inhibit, SOCE. These observations support the view that Pyr6, BTP-2, and flecainide interfere with extracellular Ca^2+^ entry through Orai1.

The mechanism by which Pyr6 inhibits the I_CRAC_ and SOCE is still unclear, although it is unlikely to prevent the STIM-dependent Orai1 channel activation process [136]. BTP-2, in turn, could target a binding site that is located on the extracellular side of the Orai1 channel pore when administered at low micromolar doses [140], as throughout our investigations [4,53]. This hypothesis is supported by the evidence that BTP-2 does not block the I_CRAC_ when intracellularly applied [141] and does not prevent the physical interaction between STIM1 and Orai1 [142], as suggested for another established SOCE inhibitor, i.e., 2-Aminoethyldiphenyl Borate [143]. Future studies are mandatory to unveil how flecainide blocks constitutive SOCE in ACM C-hMSCs. Nevertheless, the evidence presented in [4] further supports the emerging view that this pleiotropic drug targets multiple signalling pathways that are involved in ACM pathogenesis: (1) Na_V_1.5 and (2) RyR2 in cardiomyocytes, thereby rescuing the spontaneous electrophysiological events that lead to tachycardia and ventricular fibrillation; (3) SOCE in C-MSCs, thereby alleviating the aberrant remodelling of cardiac structure that predisposes the heart to generate arrhythmic events. Interestingly, we have previously shown that the β-blocker propranolol inhibits constitutive SOCE and reduces the vasculogenic activity of circulating endothelial colony forming cells in infantile hemangioma (IH) both in vitro and in children affected by IH, thereby resulting in a therapeutic benefit in IH patients [144,145]. The hypothesis that propranolol effectively targets ACM by inhibiting both β2-adrenergic receptors in sinoatrial fibres and cardiac myocytes and SOCE in C-MSCs is certainly worth of further investigation.

Notably, several Orai1 inhibitors have already entered Clinical Trials for the treatment of multiple diseases, including psoriasis (CM2489 from CalciMedica), severe acute pancreatitis (CM4620 from CalciMedica) (https://clinicaltrials.gov/ct2/show/NCT03401190 (accessed on 30 May 2023)), critical COVID-19 pneumonia (Auxora, from CalciMedica) (https://clinicaltrials.gov/ct2/show/NCT04661540 (accessed on 30 May 2023)), mild asthma (RP3128 from Rizen Pharmaceuticals) (https://clinicaltrials.gov/ct2/show/NCT02958982 (accessed on 30 May 2023)), and non-Hodgkin’s lymphoma (RP4010, also from Rizen Pharmaceuticals) (https://clinicaltrials.gov/ct2/show/NCT03119467 (accessed on 30 May 2023)) [134]. These studies confirmed that SOCE inhibitors can be administered orally, are safe, and do not induce serious adverse effects in human subjects [134]. In line with this evidence, several Food and Drug Administration (FDA)-approved drugs were recently found to inhibit both the I_CRAC_ and SOCE [146]: teriflunomide, leflunomide, roflumilast, tolvaptan, and lansoprazole. Therefore, targeting SOCE represents a promising therapeutic avenue for many diseases that still lack a valid therapeutic option, including ACM.

## 7. Future Directions

The mechanism by which flecainide inhibits SOCE requires further investigation. Likewise, the molecular determinants that increase constitutive SOCE may go beyond STIM1 overexpression and include the derangement of other regulatory factors, such as an increase in STIM1 expression or a reduction in Orai2 or Orai3 protein levels. Nevertheless, these preliminary findings strongly hint at flecainide as a pleiotropic drug that owes its efficacy to its ability to target multiple pro-arrhythmic signalling pathways in ACM. In vivo investigations will have to assess whether it does interfere with cardiac fibrofatty remodelling in transgenic mouse models of ACM. It will also be imperative to assess whether flecainide does interfere with CaMKII hyperactivation in ACM C-MSCs and, if so, to untangle the signalling pathways that are recruited by CaMKII to stimulate fibro-adipogenic differentiation. In accordance with this, Maione et al. did not assess whether inhibiting SOCE or InsP_3_Rs reduced CaMKII hyperphosphorylation in ACM C-hMSCs. Addressing this issue is mandatory to provide a clear-cut landscape of the molecular mechanisms by which an increase in constitutive SOCE promotes C-hMSC fibro-adipogenic differentiation in ACM hearts. Currently, we cannot rule out the possibility that CaMKII in ACM-hMSCs is hyperphosphorylated because of a decrease in the expression and/or activity of protein phosphatases 1 and 2A, which dephosphorylate CaMKII when [Ca^2+^]_i_ returns to the baseline [147,148]. This would in turn lead to SOCE hyperactivation and ER Ca^2+^ overload since it has long been known that CaMKII is able to stimulate SOCE [149,150,151]. However, there is no doubt that the pharmacological blockade of SOCE inhibits both spontaneous Ca^2+^ oscillations and fibro-adipogenic differentiation in ACM C-hMSCs [4]. In this view, the evidence that flecainide suppresses fibro-adipogenic differentiation paves the way for future investigations assessing whether other FDA-approved SOCE inhibitors could also be used to treat ACM by preventing fibro-fatty remodelling in C-MSCs. Finally, it could also be evaluated whether flecainide is also effective to target other disorders that are associated with SOCE up-regulation, such as severe acute pancreatitis, chronic inflammation, and cancer.

## 8. Conclusions

Flecainide is a class I_c_ AAD that proved to be effective in the management of heart rate by targeting Na_V_1.5, which mediates the fast inward Na^+^ current, and RyR2, which mediates aberrant SR Ca^2+^ release. Flecainide is emerging as a promising therapeutic tool to treat ACM, which involves a complex remodelling of the Ca^2+^-handling machinery in ventricular cardiomyocytes. In addition, studies conducted on C-MSCs isolated from ACM patients and transgenic mouse models of ACM revealed that intracellular Ca^2+^ dynamics is also dysregulated in the stromal compartment of the heart. SOCE, which represents the main intracellular Ca^2+^ entry pathway in non-excitable cells, is constitutively activated in ACM C-hMSCs and causes Ca^2+^ overload in the InsP_3_-sensitive ER Ca^2+^ reservoir. This results in an increase in both the amplitude and frequency of spontaneous Ca^2+^ oscillations that finely tune MSC fate, thereby favouring fibro-adipogenic differentiation in ACM C-hMSCs. Therefore, SOCE up-regulation in C-hMSCs could play a pathogenic role in ACM by stimulating the cardiac structural remodelling that increases the incidence of arrhythmic events. Intriguingly, flecainide, when applied at doses known to inhibit Na_v_1.2 and RyR2 channels, inhibited constitutive SOCE, thereby suppressing the spontaneous Ca^2+^ oscillations that occur during fibro-adipogenic differentiation and preventing ACM C-hMSC differentiation into fibroblasts or adipocytes. These pieces of evidence provide the first proof-of-concept that flecainide is also effective against SOCE and indicate that SOCE could be targeted for therapeutic purposes in ACM.

## Figures and Tables

**Figure 1 jcm-12-05295-f001:**
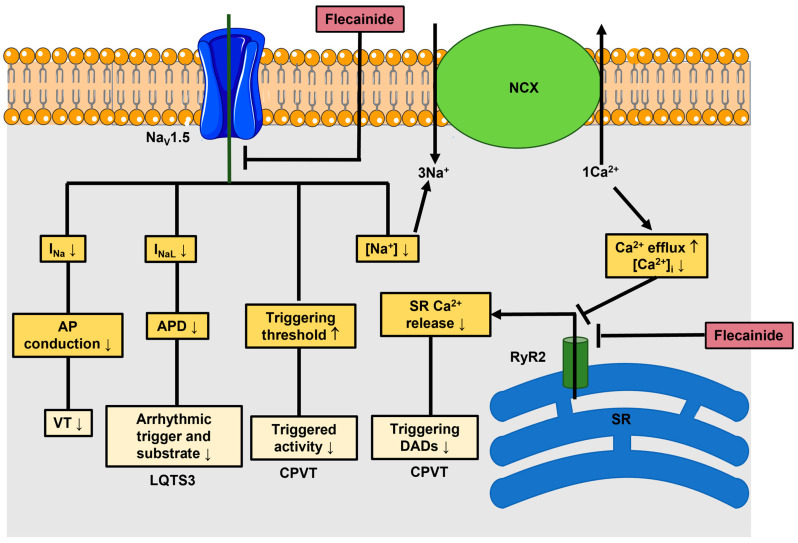
Flecainide inhibits Na_V_1.5 and RyR2. Flecainide blocks Na_V_1.5 channels, thereby blocking the fast inward Na^+^ current (I_Na_) in atrial and ventricular cardiomyocytes with a half-maximum inhibitory concentration (IC_50_) of 345 μM. By doing so, flecainide reduces the velocity of AP conduction at multiple levels: the right atrium, the atrioventricular node, and the His-Purkinje system. This, in turn, renders the heart less susceptible to VT. Flecainide-induced inhibition of I_NaL_ is also able to increase the threshold for delayed afterdepolarization (DAD) generation in response to aberrant RyR2-mediated SR Ca^2+^ release in catecholaminergic polymorphic ventricular tachycardia (CPVT). Flecainide binds to the extracellular side of the Na_v_1.5 channel pore when it opens in response to membrane depolarization and attenuates extracellular Na^+^ entry into the cytoplasm. Subsequent closing of the inactivating gate (i.e., cytosolic III–IV linker) traps flecainide within the channel pore such that flecainide-induced Na_v_1.5 inhibition increases by accelerating pulsing frequency and is, therefore, use-dependent. Under these conditions, the IC_50_ drops to 7.4 μM [22]. Flecainide can also block I_NaL_, which results from a gain-of-function mutation of Na_V_1.5, thereby reducing the AP duration (APD) in Purkinje fibres and effectively treating LQTS3. The IC_50_ of flecainide-induced inhibition of wild-type I_NaL_ was significantly lower as compared to inactivation-deficient I_NaL_ i.e., 0.61 µM vs. 365 µM [29]. RyR2 represents an additional molecular target for flecainide. Flecainide may directly inhibit RyR2, thereby normalising aberrant SR Ca^2+^ release in CPVT. In addition, by inhibiting I_Na_, flecainide may also reduce the intracellular Na^+^ concentration ([Na^+^]_i_), thereby favouring cytosolic Ca^2+^ efflux through the Na^+^/Ca^2+^ exchanger (NCX) and thus reducing Ca^2+^ sequestration into the SR. This indirectly reduces RyR2-mediated Ca^2+^ release in CPVT because of a reduction in SR free Ca^2+^ levels.

**Figure 2 jcm-12-05295-f002:**
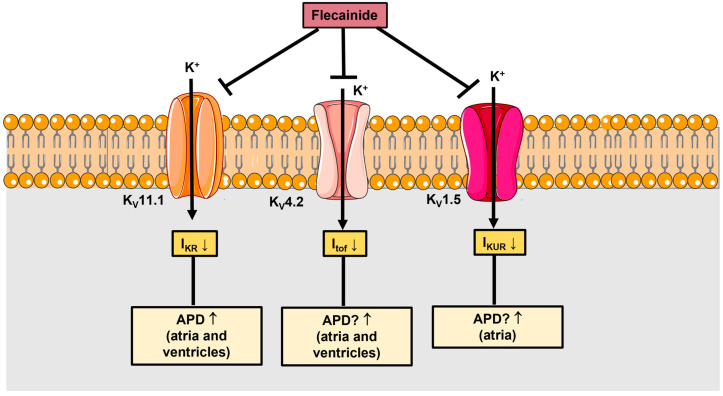
Flecainide inhibits voltage-gated K_V_ channels. At concentrations lower than 10 µM (IC_50_ = 1.5 µM), flecainide inhibits K_v_11.1, which mediates the rapid delayed rectifier K^+^ current (I_KR_), also known as hERG, thereby increasing APD in atrial and ventricular cardiomyocytes. Docking simulations indicate that flecainide targets I_KR_ by accessing the K_v_11.1 channel cavity from the cytosol and thereafter interacting with a binding site that is located low in the channel pore [31].

**Figure 3 jcm-12-05295-f003:**
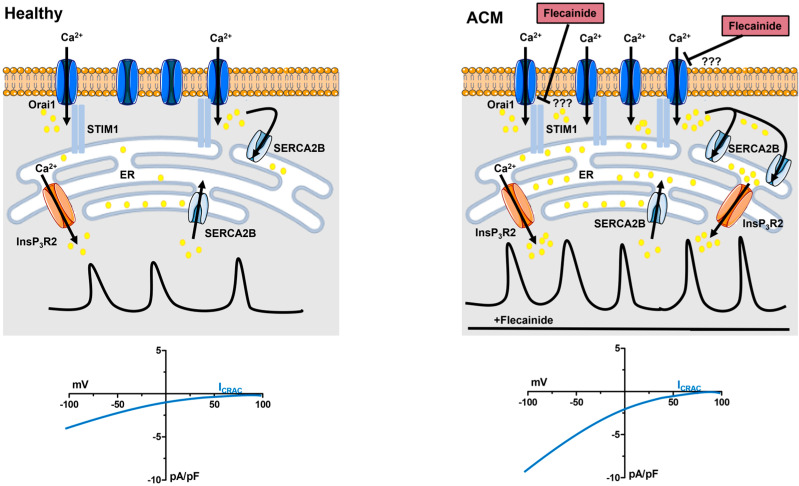
Remodelling of the Ca^2+^ handling machinery in ACM C-hMSCs. In healthy C-MSCs (C-hMSCs), constitutive SOCE is mediated by the interaction between STIM1 and Orai1, thereby replenishing the ER with Ca^2+^ in a SERCA2B-dependent manner. Periodic ER Ca^2+^ release through InsP_3_Rs, including InsP_3_R2, leads to spontaneous Ca^2+^ oscillations. The basal I_CRAC_ in C-hMSCs has yet to be measured but is likely to display the archetypal biophysical features of the I_CRAC_ originally described in mast cells, including the inwardly-rectifying current-to-voltage relationship and a reversal potential more positive than +60 mV. In ACM C-hMSCs, constitutive SOCE is enhanced due to STIM1 protein overexpression. This is likely to be reflected in an increase in the I_CRAC_ (to be experimentally demonstrated). The increase in constitutive SOCE is associated with the up-regulation of SERCA2B and InsP_3_R2 proteins. This complex remodelling of the Ca^2+^ handling machinery causes an increase in the amplitude and frequency of spontaneous Ca^2+^ oscillations in ACM C-hSMCs. Flecainide suppresses these repetitive Ca^2+^ spikes by blocking constitutive SOCE. The mechanism by which flecainide inhibits SOCE is still unclear (as denoted by ???): it could either directly plug the Orai1 channel pore or prevent the physical association between STIM1 and Orai1.

**Figure 4 jcm-12-05295-f004:**
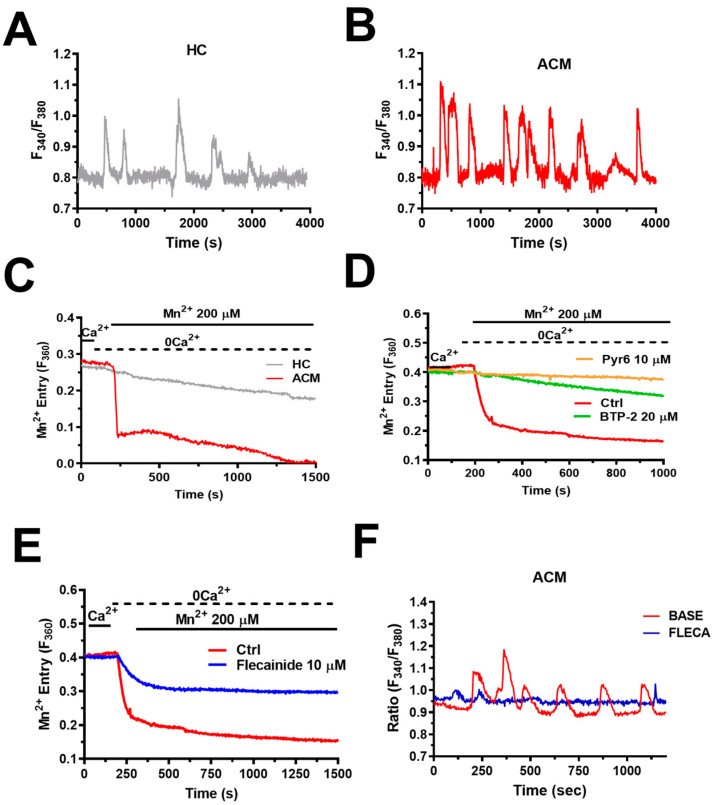
Spontaneous Ca^2+^ oscillations and constitutive SOCE are up-regulated in ACM C-hMSCs. (**A**), representative spontaneous Ca^2+^ oscillations in a C-hMSC loaded with the Ca^2+^-sensitive fluorophore, Fura-2. HC: healthy control. (**B**), representative spontaneous Ca^2+^ oscillations in an ACM C-hMSC loaded with the Ca^2+^-sensitive fluorophore, Fura-2. Note that the amplitude and frequency of repetitive Ca^2+^ spikes are enhanced as compared to C-hMSCs. ACM: arrhythmogenic cardiomyopathy. (**C**), the Mn^2+^-quenching assay technique revealed that the rate of constitutive Ca^2+^ entry was larger in ACM C-hSMCs. Most Ca^2+^-permeable channels, including Orai1, are permeable to Mn^2+^. In these experiments, extracellular Ca^2+^ is replaced by Mn^2+^ and EGTA is added to the extracellular solution to buffer any remaining Ca^2+^ trace. In the presence of Ca^2+^-permeable channels that are constitutively open on the plasma membrane, extracellular Mn^2+^ rapidly diffuses within the cytosol and quenches Fura-2 fluorescence at 360 nm, which is the isosbestic point for this ratiometric Ca^2+^ indicator. The rate of Mn^2+^-induced Fura-2 fluorescence quenching is thus indicative of the rate of basal Ca^2+^ entry. HC: healthy control. ACM: arrhythmogenic cardiomyopathy. (**D**), the Mn^2+^-quenching assay technique revealed that the enhanced constitutive SOCE in ACM C-hMSCs was inhibited by the selective Orai1 inhibitors, Pyr6 (10 µM) and BTP-2 (20 µM). (**E**), constitutive SOCE in ACM C-hMSCs was significantly (*p* < 0.05) reduced by flecainide (10 µM). Ctrl: control. (**F**), 10 µM flecainide suppressed spontaneous Ca^2+^ oscillations in ACM C-hMSCs. Base: oscillations recorded in the absence of flecainide. Fleca: oscillations recorded in the presence of 10 µM flecainide. Modified from [16] (https://creativecommons.org/licenses/by/4.0/ (accessed on 10 June 2023)).

## Data Availability

Not applicable.

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
