# Peer review of "Store-Operated Ca2+ Entry as a Putative Target of Flecainide for the Treatment of Arrhythmogenic Cardiomyopathy"

_jcm, 2023, doi:10.3390/jcm12165295_

Round 1
Reviewer 1 Report
Dr. Moccia and colleagues have described really exhaustively the role of flecainide as potential therapeutic drug for Arrhythmogenic cardiomyopathy targeting SOCE and the differentiation of cardiac mesenchymal stromal cells into fibroblast and adipocytes. However, there are some important points that use be addressed:
- When authors write about genes involved in ACM, they report that half of ACM patients present mutations in desmosomal proteins or adherence junction proteins, as cadherin 2 (CDH2) or catenin α3 (CTNNA3). This is not exactly correct, as genes definitely involved with ACM are desmosomal encoding genes and CDH2 or CTNNA3 are involved in extremally rare cases. As reported by James C. et al (DOI: 10.1161/CIRCGEN.120.003273), those genes have a limited association with ACM and I think this could be added in the review.
- The molecular mechanism of flecainide is fully reported, including data on concentration and kinetics, but I think it is to specific and some sections are difficult to follow. I think this part could be re-write with less details as it is possible to use the figures for specific information.
- Are there some side effects on using flecainide? It could be useful a small paragraph describing side effects or possible negative results of using flecainide in ACM.
- The title is quite specific and it is not clear that we are talking about a review, it seems a research article. And section 6 summarize the already published paper by Maione A.S. et al. It could be useful to add a wider point of view because it seems the continuation of the same article.
None
Author Response
We are truly grateful for your insightful comments on our manuscript entitled: “Store-operated Ca2+ entry (SOCE) as a putative target of flecainide for the treatment of arrhythmogenic cardiomyopathy” submitted for publication as Review article in Journal of Clinical Medicine.
The manuscript has been carefully revised according to your suggestions. All the changes were marked in Red. We truly believe that your comments significantly improved the quality of our work and are very thankful for this.
More specifically:
When authors write about genes involved in ACM, they report that half of ACM patients present mutations in desmosomal proteins or adherence junction proteins, as cadherin 2 (CDH2) or catenin α3 (CTNNA3). This is not exactly correct, as genes definitely involved with ACM are desmosomal encoding genes and CDH2 or CTNNA3 are involved in extremally rare cases. As reported by James C. et al (DOI: 10.1161/CIRCGEN.120.003273), those genes have a limited association with ACM and I think this could be added in the review.
We do thank the Reviewer for this insightful comment, that we have incorporated in the text. The genetic analysis by James et al. was discussed in the Introduction and lines 44-55 were reworded to fully take in account the conclusions of this study that were kindly brought to our attention by the Reviewer.
The molecular mechanism of flecainide is fully reported, including data on concentration and kinetics, but I think it is to specific and some sections are difficult to follow. I think this part could be re-write with less details as it is possible to use the figures for specific information.
We thank the Reviewer for this comment, which is quite appropriate. We had the same doubts throughout all the writing of this manuscript. The reason why we decided to add the data regarding flecainide dose and kinetics is that this manuscript was meant to be read by cardiologists and arrhythmologists, who are of course expert with flecainide treatment, and colleagues dealing with Ca2+ signalling. The latter are much less familiar with ACM and tend to associate SOCE with other diseases, such as SCID, cancer, pancreatitis, and inflammatory disorders. Therefore, we thought that they could benefit from a wider explanation of flecainide use in treating arrhythmias. Nevertheless, we amended the manuscript by following your kind suggestion. The concentration and mechanism of action (when known) of flecainide were described in Figure legends as far as it concerns voltage-gated Na+ and K+ channels. The IC50 of flecainide for RyR2 has been left in the main text since RyR2 is a crucial component of the Ca2+ handling machinery that represents the main subject of this manuscript. Of course, we might also move this information in the corresponding Figure legend should this be required by the Reviewer.
Are there some side effects on using flecainide? It could be useful a small paragraph describing side effects or possible negative results of using flecainide in ACM.
We thank the Reviewer for this observation. Actually, a number of reviews recently addressed this issue, and we quoted them in the original version of the manuscript. Indeed, in Paragraph 2, lines 185-194, we have described the potential side effects of flecainide reported by the Cardiac Arrhythmia Suppression Trial (CAST) study in patients with myocardial infarction, left ventricular dysfunction or structural heart disease. However, a series of recent investigations, that we have quoted in the manuscript, as well as a retrospective analysis by Rolland et al. (10.1093/europace/euab182), confirmed that flecainide is both effective and safe for reducing atrial fibrillation and symptomatic ventricular arrhythmias in patients with different structural heart disorders, including coronary artery disease. Thus, we believe that we have already addressed this issue in the original version of the manuscript. However, to follow the Reviewer’s suggestion, we have slightly elongated this piece of discussion, which now ends at line 202. In addition, we have now reported the main side effect reported by the CAST study, i.e., arrhythmias.
The title is quite specific and it is not clear that we are talking about a review, it seems a research article. And section 6 summarize the already published paper by Maione A.S. et al. It could be useful to add a wider point of view because it seems the continuation of the same article.
We thank the Reviewer for this comment. Actually, already in the original version of the manuscript, Section 6 addressed many issues that were not even mentioned in the original research article by Maione et al. For instance, we have now discussed the evidence that the frequency of the spontaneous Ca2+ oscillations arising in ACM C-hMSCs might not be in the most suitable range to recruit CaMKII if we do not take in account the increase in the duration of the individual Ca2+ spikes (lines 475-485). Always in the original version of the manuscript, we had highlighted a potential caveat to the conclusions of the original research article by Maione et al, i.e., the lack of any evidence that blocking SOCE (and therefore the spontaneous Ca2+ oscillations) reduces CaMKII hyperphosphorylation (while blocking SOCE always prevents fibro-fatty differentiation). Finally, in the original version of the manuscript, we had discussed the possibility that several FDA-approved CRAC inhibitors, including propranolol, could be tested against ACM (lines 538-558).
However, to address the kind concern raised by the Reviewer, we have further expanded Section 6 by discussing additional points: 1) the potential contribution of lysosomal Ca2+ release to the spontaneous Ca2+ oscillations driving ACM C-hMSC fibro-adipogenic differentiation (lines 428-444); 2) an hypothesis about the mechanism by which CaMKII could stimulate fibro-adipogenic differentiation (lines 485-490); and 3) a novel Paragraph entitled “Future directions”, in which we moved some of the concepts originally presented in the Conclusions (that were shortened) and discussed about the possibility that CaMKII hyperphosphorylation rather reflects a decrease in the expression and/or activity of the protein phosphatases 1 and 2a that dephosphorylate CaMKII when the [Ca2+]i returns to the baseline.
We, therefore, hope that this article will be suitable for this exciting Special Issue of Journal of Clinical Medicine.
Reviewer 2 Report
The authors conducted a comprehensive review on Store-operated Ca2+ entry (SOCE) as a putative target of flecainide for the treatment of arrhythmogenic cardiomyopathy. In general, the review is thorough, although, this being highly "clinical" topic, I wonder why the authors did not expand the part concerning the change in paradigm with regards to flecainide, as well as clinical data to support it. I think that this aspect is valuable to address, especially since the review is concerning putative mechanisms.
Adding some details about pathophysiological background of ARVC as a separate paragraph would perhaps be beneficial.
Minor editing of English language required.
Author Response
We are truly grateful for your insightful comments on our manuscript entitled: “Store-operated Ca2+ entry (SOCE) as a putative target of flecainide for the treatment of arrhythmogenic cardiomyopathy” submitted for publication as Review article in Journal of Clinical Medicine.
The manuscript has been carefully revised according to your suggestions. All the changes were marked in Red. We truly believe that your comments significantly improved the quality of our work and are very thankful for this.
More specifically:
Adding some details about pathophysiological background of ARVC as a separate paragraph would perhaps be beneficial.
We thank the Reviewer for this comment. The Introduction has been shortened and an entire paragraph, i.e., Paragraph 2, has been devoted to describe the pathological background of ACM and the most common therapeutic options for this disease.
Minor editing of English language required.
We do thank the Reviewer for this comment. The manuscript has been carefully revised.
We, therefore, hope that this article will be suitable for this exciting Special Issue of Journal of Clinical Medicine.

Reviewer 3 Report
By this manuscript Author gave an an overview of the different moleculr aspects of ACM focusing on the description of flecanide effects/interactions hence indicating this molecule as a potential drug to be employed in arrythmogenic cardiomyopathy theraphy.
Review is nicely organized, fluent and clear and all the topics are deeply investigated and discussed.
No concerns from my side but please revise paragraph numbering since paragraph 2 is missing
Author Response
We are truly grateful for your nice comments on our manuscript entitled: “Store-operated Ca2+ entry (SOCE) as a putative target of flecainide for the treatment of arrhythmogenic cardiomyopathy” submitted for publication as Review article in Journal of Clinical Medicine.
We have revised paragraph numbering and truly thank you for noticing our mistake.
We, therefore, hope that this article will be suitable for this exciting Special Issue of Journal of Clinical Medicine.
Reviewer 4 Report
I enjoyed reading the present review article from Moccia and colleagues. The Authors provide a very good coverage of recent acquisitions on the physiopathology and treatment of arrhythmogenic cardiomyopathy (ACM). More specifically, they discuss, in great detail, the primary molecular mechanisms targeted by flecainide to mitigate arrhythmic events in ACM, including Nav1.5 and RyR2. They describe the role of spontaneous Ca2+ oscillations in determining cardiac mesenchymal stromal cells (C-MSCs) fate, and how they do contribute to ACM pathophysiology by differentiating into fibroblasts and adipocytes. Then, they tackle recent work dealing with spontaneous Ca2+ oscillations in ACM. Finally, they discuss how flecainide can suppress spontaneous Ca2+ oscillations and fibro-adipogenic differentiation in ACM C-hMSCs by inhibiting constitutive SOCE.
Current review is of great interest and very helpful to readers, expert or not in ACM.
I have only few minor comments to make.
1) Introduction (lines 55-57). When considering the contribution of desmosome alterations, recent evidence shows that calcium overload in Dsg2 mut/mut hearts (subjected to exercise or an ISO challenge) induces calpain-1 activation that, in turn, prompts the cleavage of mitochondrial-bound apoptosis-inducing factor (AIF); ultimately, truncated (and oxidized) AIF migrates to the nucleus when it triggers myocyte death (Chelko SP. et al., Science Trans Med., 2021). It would nice to include this recent evidence because it shows a causal link between Ca2+ mishandling and myocyte death in ACM.
2) I suggest the Authors to keep the Conclusion paragraph very short. Perhaps, it can be preceded by an additional paragraph entitled Studies in Perspective (or something like that) in which the detail the ongoing new directions and/or needs in the ACM pathophysiology/treatment field.
3) A final cartoon that summarizes the main mechanisms discussed/possible therapeutic targets would help the readers. Just a suggestion.
Author Response
We are truly grateful for your insightful comments on our manuscript entitled: “Store-operated Ca2+ entry (SOCE) as a putative target of flecainide for the treatment of arrhythmogenic cardiomyopathy” submitted for publication as Review article in Journal of Clinical Medicine.
The manuscript has been carefully revised according to your suggestions. All the changes were marked in Red. We truly believe that your comments significantly improved the quality of our work and are very thankful for this.
More specifically:
1) Introduction (lines 55-57). When considering the contribution of desmosome alterations, recent evidence shows that calcium overload in Dsg2 mut/mut hearts (subjected to exercise or an ISO challenge) induces calpain-1 activation that, in turn, prompts the cleavage of mitochondrial-bound apoptosis-inducing factor (AIF); ultimately, truncated (and oxidized) AIF migrates to the nucleus when it triggers myocyte death (Chelko SP. et al., Science Trans Med., 2021). It would nice to include this recent evidence because it shows a causal link between Ca2+ mishandling and myocyte death in ACM.
We do thank the Reviewer for suggesting mentioning this remarkable article in Paragraph 4, lines 232-239, which specifically deals with the remodelling of the Ca2+ handling machinery in ACM.
2) I suggest the Authors to keep the Conclusion paragraph very short. Perhaps, it can be preceded by an additional paragraph entitled Studies in Perspective (or something like that) in which the detail the ongoing new directions and/or needs in the ACM pathophysiology/treatment field.
We do thank the Reviewer for this suggestion. We shortened the Conclusions and added a preceding paragraph entitled: Future directions. Herein, we moved some of the observations that we initially included in the Concluded and provided some more suggestions regarding the future research in the field of ACM. The Reviewer was quite right. The Conclusions were too long!
3) A final cartoon that summarizes the main mechanisms discussed/possible therapeutic targets would help the readers. Just a suggestion.
We thank the Reviewer for this suggestion. Figure 3 represents the main signalling pathways discussed in the study and highlights the putative targets of flecainide in ACM C-hMSCs. In order to fully address the kind suggestion by the Reviewer, we have added a flat Ca2+ trace in the panel showing the remodelling of the Ca2+ signalling machinery in ACM C-hMSCs. This flat Ca2+ trace represents the consequences on spontaneous Ca2+ oscillation of flecainide-induced inhibition of constitutive SOCE.
We, therefore, hope that this article will be suitable for this exciting Special Issue of Journal of Clinical Medicine.